# Is the HTLV-1 Retrovirus Targeted by Host Restriction Factors?

**DOI:** 10.3390/v14081611

**Published:** 2022-07-23

**Authors:** Auriane Carcone, Chloé Journo, Hélène Dutartre

**Affiliations:** CIRI—Centre International de Recherche en Infectiologie, Université Claude Bernard Lyon 1, Inserm, U1111, CNRS, UMR5308, ENS Lyon, 69007 Lyon, France; auriane.carcone@ens-lyon.fr (A.C.); chloe.journo@ens-lyon.fr (C.J.)

**Keywords:** human T cell leukemia virus type 1, restriction factors, intrinsic immunity, viral replication cycle

## Abstract

Human T cell leukemia virus type 1 (HTLV-1), the etiological agent of adult T cell leukemia/lymphoma (ATLL) and of HTLV-1-associated myelopathy/tropical spastic paraparesis (HAM/TSP), was identified a few years before Human Immunodeficiency Virus (HIV). However, forty years later, our comprehension of HTLV-1 immune detection and the host immune responses to HTLV-1 is far more limited than for HIV. In addition to innate and adaptive immune responses that rely on specialized cells of the immune system, host cells may also express a range of antiviral factors that inhibit viral replication at different stages of the cycle, in a cell-autonomous manner. Multiple antiviral factors allowing such an intrinsic immunity have been primarily and extensively described in the context HIV infection. Here, we provide an overview of whether known HIV restriction factors might act on HTLV-1 replication. Interestingly, many of them do not exert any antiviral activity against HTLV-1, and we discuss viral replication cycle specificities that could account for these differences. Finally, we highlight future research directions that could help to identify antiviral factors specific to HTLV-1.

## 1. Introduction

Human T cell leukemia virus type 1 (HTLV-1) is a Deltaretrovirus estimated to chronically infect 5–10 million people worldwide, a number that is probably underestimated [1]. The majority of people living with HTLV-1 remain life-long asymptomatic carriers. However, in 5% of the cases, infection leads to the development of one of the following severe pathologies: Adult T cell leukemia/lymphoma (ATLL), or HTLV-1 associated myelopathy, also called tropical spastic paraparesis (HAM/TSP) [2,3]. In addition, HTLV-1 is also the etiologic agent of uveitis and infectious dermatitis and has been associated with strongyloidiasis [4]. At present, the physio-pathological mechanisms governing the progression towards one or another of these pathologies remain largely unknown, and there is no satisfying treatment available.

Upon chronic infection, the major cellular target of HTLV-1 is CD4+ T cells, in which the viral oncogenic process can result in the development of leukemia/lymphoma. Other immune cells can also be infected, such as CD8+ T cells and myeloid cells. In particular, dendritic cells (DC) have been suggested to act as the first cells encountered during primo-infection [5]. DC can be productively infected by HTLV-1 and are able to transmit the virus to CD4+ T cells [6,7,8]. However, in vitro studies in immature DC demonstrated that despite their high capacity to capture the virus, only a small proportion of them are productively infected and able to support a complete viral replication cycle [7]. Although this might be sufficient for establishing chronic infection, this suggests that the viral replication cycle is compromised in the majority of susceptible DC. In addition, mature DC are completely resistant to infection [8]. The mechanisms accounting for this poor infectivity of HTLV-1 in DC are not fully understood, but could be linked to intrinsic defense mechanisms in these host cells.

As a retrovirus, the HTLV-1 genome is integrated in the host cell genome. HTLV-1 chronic infection has been shown to be maintained by clonal proliferation of infected CD4+ T cells (Figure 1) [9]. This strategy allows for viral dissemination without the need for the production and release of new viral particles, and thus, reduces both the genetic variability of the virus, since reverse transcription is not used, and the exposure of the virus to possible antiviral factors. Nonetheless, it is now largely admitted that HTLV-1 chronic infection is also maintained by continuous infectious viral spread [10,11]. Intriguingly, HTLV-1-infected cells release low numbers of viral particles that are poorly infectious [12,13]. Accordingly, HTLV-1 infectious spread is thought to occur by cell-to-cell transfer using a biofilm (defined as an accumulation of viral particles associated with the extracellular matrix at the surface of chronically infected T cells), viral synapses, or protrusions and tunneling nanotubes [14], rather than by cell-free transmission (Figure 1). Strikingly, the proviral load, reflecting a successful dissemination of HTLV-1 both by clonal expansion of infected cells and by infectious spread, is very low in vivo, even in CD4+ T cells [15]. Combined with the observation that de novo infection of primary CD4+ T cells is also poorly efficient in vitro even through cell-to-cell contacts [7], this further strengthens the notion that host factors could exert an intrinsic block to HTLV-1 infection.

In addition to innate and adaptive immune responses that rely on specialized cells of the immune system, host cells may also express a range of antiviral factors that inhibit viral replication at different stages of the cycle, in a cell-autonomous manner. These antiviral factors can be constitutively expressed, and therefore act as immediate intrinsic barriers to viral replication. However, most of the antiviral factors are inducible upon the innate sensing of viral infection by pattern recognition receptors, as they are encoded by type-I interferon (IFN-I)-stimulated genes (ISG). This feature of antiviral factors thus bridges intrinsic and innate immunity, as IFN-I can be produced in large amounts by specialized innate cells such as plasmacytoid dendritic cells (pDC) and myeloid cells.

Cell-autonomous antiviral factors were originally discovered in the HIV field, and multiple HIV-directed antiviral factors have been identified since then, belonging to diverse protein families and exerting antiviral activities through diverse mechanisms [16,17]. A subclass of these antiviral factors is referred to as “restriction factors” by some authors. These should meet a number of specific criteria, including being essentially ineffective against wild-type viruses infecting their natural host because of viral escape mechanisms or viral antagonism [18]. Antiviral factors for which no escape mechanisms or viral antagonism have been identified are dubbed “resistance factors” instead [18]. However, these strict definitions are debated, and other authors use the term “restriction factor” in a broader sense, to include all cellular factors with known antiviral activities that manifest in a cell-autonomous manner [19]. This broad acceptation of the term will be used hereafter.

Being close to the lentivirus HIV in terms of viral organization, tropism and viral replication cycle, it is tempting to speculate that HTLV-1, although belonging to a different retroviral genus, might also be controlled by restriction factors with demonstrated antiviral activity against HIV. However, while the infection of CD4+ T cells by HIV leads to the massive release of neo-synthesized viral particles, and to the death of the infected cell, infection by HTLV-1 leads to the rapid shut-down of viral particle production, and induces CD4+ T cell spontaneous proliferation that can progress to cell immortalization and transformation. Hence, while host T cells might be under a strong selection pressure to restrict HIV replication cycle, this might not be applicable to host T cells exposed to HTLV-1. In this work, we review the extent to which HIV restriction factors can restrict HTLV-1 replication (Figure 2). As expected from the different outcomes of HIV or HTLV-1 infections, many of these are found to be inactive against HTLV-1. Therefore, we discuss differences between HTLV-1 and HIV replication cycles that could account for these specificities. This comparative analysis leaves open the question of how HTLV-1 replication is controlled in susceptible cells, resulting in such a low yield of infected cells. We conclude this review article by highlighting future directions in the HTLV-1 basic research field that may foster the discovery of new restriction factors active against HTLV-1.

## 2. HTLV-1 Is Poorly Sensitive to Type-I Interferon, a Potent Inducer of HIV Restriction Factors

As stated in the introduction, most HIV restriction factors are encoded by ISG [20,21], explaining the long-standing observation that IFN-I potently inhibits HIV replication in cell cultures. The effects of IFN-I treatment on HTLV-1 infection have thus been thoroughly investigated.

Most studies focused on the possible activity of IFN-I in the late stages of the viral replication cycle and in the proliferation of chronically infected T cells. In several in vitro studies, Kinpara et al. demonstrated that an IFN-α treatment suppressed HTLV-1 gene expression and the proliferation of HTLV-1 infected T cells [22,23]. Similarly, peripheral blood mononuclear cells (PBMC) from ATL patients treated ex vivo with IFN-α displayed decreased viral expression and cell proliferation, as well as increased apoptosis [24], indicating that the late steps of the HTLV-1 replication cycle, and in particular the step of viral gene expression, are sensitive to the antiviral activity of IFN-I. We also confirmed these results in de novo infected T cell lines and primary T cells, in which IFN-α restricted the late stages of the viral replication cycle [25]. Of note, IFN-α had no inhibitory effect on the early stages of HTLV-1 infection [25]. This is different from HIV, as IFN-I induces several restriction factors that target both early and late stages of HIV replication cycle [26].

We further showed that IFN-I-induced restriction of the late stages of HTLV-1 infection in T cells is mediated by Protein Kinase R (PKR), whose activation reduces the translation of HTLV-1 mRNA [25]. PKR also prevents HIV mRNA translation, as well as that of multiple viral families [25,27]. While HIV Tat protein was shown to antagonize PKR activity [28], no direct HTLV-1 antagonism of PKR activity has been described so far. Surprisingly, however, we identified ADAR1 (Adenosine Deaminase Acting on RNA) as an ISG that exerts a proviral activity on HTLV-1 in T cells, by inhibiting PKR [29]. Thus, upon IFN-I treatment, the fine balance between the induction of anti- and proviral factors might determine the net effect of IFN-I observed in the viral replication cycle, independent of a specific antagonism provided by viral proteins

Importantly, IFN-I might differentially affect HTLV-1 infection depending on the cell type. Accordingly, the HTLV-1 auxiliary p30 protein is required for the infection of monocytes and dendritic cells, but not of T cells [30]. This indicates that p30 might overcome a block in monocytes and dendritic cells, possibly by dampening antiviral ISG expression [31,32], and that this block would not be active in T cells. However, the exact nature of this block remains to be identified. In addition, in contrast to its effect on T cells, IFN-I treatment of DC prior to their exposure to HTLV-1 did not reduce their infection nor viral expression [8], suggesting that PKR might not be active in this context. Whether this is linked to a specific balance between PKR and ADAR1 expression induced by IFN-I treatment in DC, and/or to an HTLV-1 antagonism that would be restricted to this cell type, remains to be investigated.

## 3. Activity of Known HIV Restriction Factors on HTLV-1 Replication

Until now, no unbiased systematic screen for HTLV-1 restriction factors has been performed. Instead, most studies have analyzed the ability of known HIV restriction factors to restrict HTLV-1 replication. Because HTLV-1 shares a similar replication cycle with HIV, it is indeed tempting to hypothesize that HIV restriction factors are also active against HTLV-1. However, one should bear in mind that our understanding of some of the steps of the HTLV-1 replication cycle has been extrapolated from the HIV field, with no confirmation of the specific context of HTLV-1 infection. Therefore, the actual extent of similarity between the cycles of both viruses, in their molecular details, remains hypothetical. This holds true in particular for the early steps of uncoating, reverse transcription, nuclear transport and integration [33]. In contrast, the expression and function of regulatory and auxiliary proteins encoded by the pX region of HTLV-1, among which Tax, HBZ, and p30, have been extensively studied. Although HTLV-1 Tax shares some features with HIV Tat, such as its function as a viral transactivator, the respective functions of HTLV-1 and HIV regulatory and auxiliary proteins are mostly unrelated, with Tax and HBZ showing the unique ability to promote T cell proliferation, a process that can pave the way for cell immortalization and transformation.

Indeed, as stated above, infection by HTLV-1 induces persistent T cell polyclonal infected cells proliferation, a process that has been observed both in vitro and in vivo, and independent of the clinical status of the infected individual (i.e., in asymptomatic carriers, HAM/TSP patients as well as ATL patients) [34]. Seminal research by Eric Wattel and colleagues, using the ex vivo culture of single isolated cells from infected individuals without malignancy, has shown that the clonal expansion of infected CD4+ T cell primarily results from HTLV-1 propelling the host cell into the cell cycle [35] in a Tax-dependent manner. Tax indeed exerts multiple actions on the cell cycle, for instance by inhibiting negative regulators such as p53, p16^INK4A^ and p27^Kip1^, and by activating positive regulators such as CDK4/6 and E2F (for a review, see [36]). HBZ was also shown to enhance cell proliferation, by acting both in its mRNA and protein forms [37,38]. Tax and HBZ also contribute to cell immortalization, by reactivating the expression of the human telomerase reverse transcriptase (hTERT) gene, and modulating the activity of the telomerase complex [39,40] Importantly, while HTLV-1 regulatory proteins are necessary to induce the initial steps of cell proliferation and immortalization, ultimately leading to cell transformation, infected cells might acquire a “HTLV-1 memory”, even in the absence of viral gene expression [41]. This occurs in particular through the acquisition of specific epigenetic signatures initiated by Tax and HBZ interaction with histone-modifying enzymes such as EZH2, SUV39H1, HDAC1, and the SWI/SNF chromatin remodeling family, and of somatic mutations allowed by the high genomic instability induced by Tax and HBZ expression (recently reviewed in [42]). Nonetheless, these effects of Tax and HBZ on cell proliferation occur during the late stages of viral replication, and thus after viral entry, uncoating, reverse transcription and integration. Those early stages might be consequently susceptible to cellular antiviral responses that would prevent the infection of target cells.

Importantly, several auxiliary proteins from HIV have demonstrated roles as antagonists of restriction factors, such as Nef or Vif, but these functions have not been identified in HTLV-1 proteins yet. This highlights the fact that the selection pressure exerted by host factors on both viruses might be very different, as will be discussed later in this review. In this section, we review the action or lack of action on HTLV-1 of some of the known HIV restriction factors, with a focus on TRIM family members, APOBEC3, SAMHD1, CIITA, ZAP and Tetherin (Table 1). We further discuss how these observations could be linked to differences in the viral replication cycle of both viruses.

### 3.1. Factors Targeting Viral Uncoating: TRIM5α

The tripartite motif (TRIM) family contains many interferon-induced proteins involved in restriction of several viruses, including lentiviruses [60]. Among them, TRIM5α targets the early stages of HIV infection after cell entry, through binding to the HIV capsid [61] and forming a cage around the capsid that impairs later uncoating [62] and/or directs capsids for degradation after their ubiquitination [63]. No HIV proteins have been described to directly counteract the action of TRIM5α, and resistance to this restriction factor relies on capsid mutation at the cost of viral fitness [64].

No direct interaction of TRIM5α with HTLV-1 capsid has been reported thus far, but one study performed on PBMC from HAM/TSP patients reported an association of TRIM5α polymorphisms with high proviral loads, suggesting that TRIM5α could also control HTLV-1 replication [43]. This result was later confirmed in a transcriptomic study conducted on PBMC from asymptomatic and HAM/TSP patients, in which all infected individuals displayed a negative correlation between TRIM5α expression and HTLV-1 infection (assessed by the proviral load and the levels of Tax and HBZ mRNA) [44]. While these correlations suggest that TRIM5α may restrict HTLV-1, direct evidence is still lacking, as are the mechanisms of this possible restriction. Of note, uncoating is one of the most poorly described steps of the HTLV-1 replication cycle, and a better characterization of this step might help to better understand the possible restriction of HTLV-1 by TRIM5α.

Following these findings, other TRIM family members were investigated, and among them, TRIM19, also known as PML, was demonstrated to restrict HTLV-1 [45]. This restriction involves Tax SUMOylation followed by its poly-ubiquitination, two modifications that ultimately lead to Tax proteasomal degradation, and thus to a lower viral expression. Interestingly, this restriction was observed in chronically infected T cell lines in response to arsenic/interferon therapy [45]. Other TRIM family members remain to be investigated such as TRIM22 for example, which acts as restriction factor for many viruses including HIV [65], with only one viral antagonist identified [66]. Therefore, TRIM proteins might be promising candidates when searching for cellular factors able to restrict HTLV-1 replication.

### 3.2. Factors Targeting Viral DNA Synthesis: APOBEC3 and SAMHD1

APOBEC3 proteins (A3) are a family of DNA cytidine deaminases (A3A to A3H), some of which display potent antiviral activity against HIV. Among them, A3G was the first restriction factor discovered for HIV. It is incorporated into the newly produced viral particles, and thus delivered to the target cell together with the viral genome. A3 act during reverse transcription where they induce G-to-A mutations in the viral DNA genome, thus affecting its integrity. Of note, A3 restriction may also occur without deaminase activity [67], by direct binding to the reverse transcriptase and the inhibition of RT-catalyzed elongation [68,69]. The A3G restriction of HIV is counteracted by the viral protein Vif [70,71], either by inducing its degradation or by preventing its packaging into virions [72]. 

Interestingly, although it is incorporated into HTLV-1 particles, A3G has no noticeable effect on HTLV-1 infectivity. Indeed, in vitro assays using HTLV-1 virion produced in the presence of ectopic A3G were capable of infecting T cell lines as the A3G-free virions and no G-to-A mutations were observed in the genome of cells infected with A3G-containing virions [48,51,52,73]. Strikingly, in vivo, A3G expression is increased in HTLV-1 patients [46,47], and there is evidence of an imprint of APOBEC3 selection on the viral genome [49,50], as suggested by the presence of G-to-A substitution in PBMC from asymptomatic or ATL patients [49]. However, A3G expression in HTLV-1 patients correlated neither with the clinical status nor with the proviral load [46,47]. Nonetheless, while HTLV-1 does not encode for any analog to Vif, a portion of the nucleocapsid (NC) was reported to exclude A3G from packaging [52], suggesting that at least in vitro, A3G or other A3 proteins might have some restriction activity against HTLV-1 that would be counteracted by NC. Indeed, unbiased testing of all the seven A3 cytidine deaminases ectopically expressed in HTLV-1-producing cells identified A3A, A3B and A3H hapII as potent inhibitors of HTLV-1 infectivity packaged into HTLV-1 virions [53], while A3G was barely active, as previously reported. Interestingly, although both HIV and HTLV-1 were shown to be sensitive to A3B and A3H hapII in this study, HIV was resistant to A3A restriction. Moreover, the deaminase catalytic activities of A3H hapII were not necessary for HTLV-1 restriction, while they were necessary for HIV restriction [53]. This further highlights the specific action of restriction factors in closely related retroviruses, and the importance of unbiased screening for factors specifically active in HTLV-1. 

SAMHD1 is another HIV restriction factor that targets pre-integration steps of the viral replication cycle in cells from the myeloid lineage. SAMHD1 is antagonized by the HIV-2-encoded protein Vpx [20,74], which, again, has no analog in HTLV-1 [54]. SAMHD1 has been shown to decrease the pool of available dNTP, thus inhibiting HIV reverse transcription. 

The effect of SAMHD1 on HTLV-1 is less clear. In DC and macrophages cultured in vitro, the degradation of SAMHD1 by SIV-Vpx delivered in target cells by lentivectors prior to infection did not affect HTLV-1 infection, while it efficiently increased that of HIV, suggesting that SAMHD1 is not active against HTLV-1 in DC and macrophages [54]. In contrast, in vitro experiments in primary monocytes showed that the expression of SAMHD1 induced an abortive infection by HTLV-1, which then triggered a STING-dependent apoptosis due to the accumulation of DNA replication intermediates [55]. Accordingly, the extinction of SAMHD1 in monocytes, or complementation with exogenous dNTP counteracting SAMHD1 restriction on reverse transcription, restored a complete reverse transcription, without induction of apoptosis and with HTLV-1 viral DNA detected in the nucleus of infected cells [55]. This suggests that SAMHD1 is able to restrict the replication of HTLV-1 in monocytes in vitro. Altogether, this shows that the SAMHD1 restriction of HTLV-1 might be dependent of the myeloid cell type, and thus, be less broad than that of HIV.

However, in vivo monocytes from HAM/TSP patients or asymptomatic carriers are positive for integrated viral DNA, despite high expression of SAMHD1 [75]. This could indicate a lack of SAMHD1 restriction in monocytes in vivo. Alternatively, the presence of HTLV-1 viral DNA in the genome of monocytes could result from the differentiation of infected hematopoietic stem cells (HSC) [76]. In this scenario, the reverse transcription step of the viral replication cycle would occur before the differentiation into monocytes, and thus before the expression of SAMHD1, accounting for the apparent lack of SAMHD1 restriction in monocytes.

Interestingly, the simian foamy virus (FV), a retrovirus from the Spumavirus genus, is also resistant to A3G [77] and SAMHD1 restriction [54], although it is sensitive to IFN-I [78], suggesting again, a specific action of antiviral factors to restrict different retroviruses. Interestingly, around 20% of viral particles released by FV-infected cells contain viral DNA instead of the expected viral RNA, suggesting that FV reverse transcription starts in the producer cell before the formation of the virions, and not in the target cell as expected [79]. This process could explain FV resistance to both A3G and SAMHD1, as both of these restriction factors interfere with the reverse transcription step in the target cell. Strikingly, some reports highlighted the presence of viral DNA in HIV virions, although in a very small proportion and with no impact on infectivity [80,81,82]. The presence of viral DNA in HTLV-1 particles has not been addressed yet, but if demonstrated, it could explain the resistance of HTLV-1 to A3G and SAMHD1 restriction. Again, the reverse transcription step of the HTLV-1 replication cycle is poorly described, and its in-depth analysis would help to better appreciate its possible inhibition by additional restriction factors that are yet to be identified.

### 3.3. Factors Targeting Viral Expression: CIITA and ZAP

MHC Class II Transactivator CIITA presents a particular interest in the field of anti-viral immunity as it may affect viral infection through two main activities. First, CIITA positively controls the expression of Major Histocompatibility Class II (MHC-II) on antigen presenting cells [83], controlling antigen presentation and the activation of CD4+ T cells. Thus, CIITA allows an indirect antiviral action relying on the induction of the adaptive immune response. Second, CIITA exerts a direct antiviral activity by targeting viral expression, in a cell-autonomous manner. CIITA was first described to restrict HIV infection in human T cells through the inhibition of Tat-dependent LTR transactivation [84]. This restriction was then further demonstrated in a promonocytic cell line, broadening HIV restriction by CIITA to the myeloid lineage [85,86]. Interestingly, the authors found that in myeloid cells, CIITA, TRIM22 and TRIM19/PML were recruited together into nuclear bodies during HIV infection, suggesting a collaborative action of these restriction factors in the infected cells [86].

The CIITA restriction of Deltaretroviruses was first investigated on HTLV-2, for which CIITA was demonstrated to exert antiviral activity in B and T cells [87]. Similar to what was observed for HIV, CIITA was able to inhibit the viral transactivation function of Tax-2, thus restricting viral expression [88]. Later on, a similar effect of CIITA was demonstrated in HTLV-1 [89], with an inhibition of Tax-dependent HTLV-1 LTR transactivation. Moreover, the interaction of CIITA with Tax blocked the Tax-induced activation of the NF-κB pathway [57], a crucial pathway involved in HTLV-1 induced cellular transformation. Thus, the CIITA restriction of Tax-induced viral transactivation and activation of cellular pathways could both inhibit viral replication and counteract the first steps of the oncogenic process triggered in HTLV-1 infected cells.

These findings were observed following the ectopic expression of CIITA in 293T cells, but also in a promonocytic cell line, in which expression of CIITA is constitutive, strengthening its role in a more physiologic model. However, more research is needed to investigate whether CIITA is efficient in other myeloid cells, specifically in primary dendritic cells or monocytes that are most relevant to HTLV-1 infection.

HIV, but also other retroviruses such as murine leukemia virus (MLV) or avian leukosis virus (ALV) [90,91,92], and many other viruses from unrelated families, such as alphaviruses, filoviruses and HBV, are restricted by the ZAP protein (CCCH type zinc finger antiviral protein). ZAP selectively binds to CpG-rich RNA sequences that are present in viral mRNA and that discriminate non-self mRNA from host mRNA [93]. The binding of ZAP induces the processing and degradation of these viral mRNA, thus restricting viral expression.

Interestingly, HTLV-1 transcripts were found to be particularly enriched in CpG sequences susceptible to ZAP-mediated suppression [58]. In addition, the over-expression and silencing experiments conducted in HTLV-1-infected cell lines confirmed that ZAP restricts HTLV-1 expression and production [58]. Of note, CpG-rich sequences were also found in the genomes of STLV, the simian HTLV counterpart, and in those of bovine leukemia virus (BLV), another member of the Deltaretrovirus genus, suggesting that ZAP could be a common Deltaretrovirus restriction factor [58].

### 3.4. Factors Targeting Viral Release and Viral Spread: Tetherin

Tetherin, also known as BST-2 (Bone Marrow Stromal Cell antigen 2), is a restriction factor active in the late stages of HIV infection. Tetherin inhibits viral particle release from the infected cell by retaining the particles at the cell surface [21,94]. Once again, while HIV expresses the viral protein Vpu to counteract Tetherin antiviral action [95], HTLV-1 lacks the expression of a viral Tetherin antagonist, and Tetherin expression is even higher in HTLV-1-infected CD4+ T cell lines than in non-infected cell lines. This indicates that HTLV-1 does not inhibit Tetherin expression or activity, which in turn suggests that HTLV-1 might not be affected by Tetherin restriction [59].

Importantly, the inability of Tetherin to restrict HTLV-1 could be well-explained by differences in the modes of dissemination of HTLV-1 versus HIV. While HIV efficiently uses both cell-free and cell-to-cell transmission, cell-free transmission does not significantly contribute to HTLV-1 dissemination, as stated above (Figure 1). Because Tetherin activity consists of retaining virions associated with the donor cell surface, and because only cell-free, but not cell-to-cell transmission, requires the actual dissociation of virions from the donor cell surface, cell-to-cell transmission might represent a strategy to overcome Tetherin restriction. This notion is supported by data from the bovine retrovirus field, that indicate that the bovine ortholog of Tetherin restricts the release of BLV [96], of bovine foamy virus and of bovine immunodeficiency virus (BIV) [97], but has no effect on their cell-to-cell transmission [97].

In vitro, HTLV-1 cell-to-cell transmission is highly efficient when the virus is present at the cell surface of the donor cell and concentrated in the form of the viral biofilm. Interestingly, Tetherin has been identified as a component of HTLV-1 viral biofilm [98], leading to the intriguing hypothesis that Tetherin could in fact favor HTLV-1 transmission and thus, act as a proviral factor. Taken together, the specific role of Tetherin in viral transmission mediated by viral biofilms, but also its roles in transfer by viral synapses or protrusions and tunneling nanotubes, remains to be investigated.

Overall, these observations indicate that the antiviral activity of HIV restriction factors cannot be systematically extrapolated to HTLV-1, as some of them (such as A3G and Tetherin) fail to exert any antiviral activity against HTLV-1. In the absence of any known HTLV-1-encoded antagonists of these restriction factors, this lack of activity against HTLV-1 indicates that these restriction factors are not relevant in the HTLV-1 context. While the lack of activity of some restriction factors can be clearly explained by differences in the viral replication cycle of both viruses, as is the case of Tetherin and cell-free versus cell-to-cell viral transfer, the molecular basis for the lack of activity of other factors is still poorly understood. A better characterization of the HTLV-1 replication cycle, and in particular of the early steps that have not been specifically dissected, such as uncoating, reverse transcription, nuclear import and integration, might help to better understand the contrasting activity of restriction factors on both viruses.

## 4. How Can We Further Expand Our Understanding of HTLV-1 Restriction?

### 4.1. By Increasing Our Knowledge on HTLV-1 Sensing and Induction of Restriction Factors upon Dendritic Cell Maturation

As stated above, antiviral restriction is intimately linked to innate immunity, as most restriction factors are induced by IFN-I upon viral sensing [99]. Several cellular proteins interacting with HTLV-1 RNA or DNA have been reported as innate sensors in different cell types. HTLV-1 RNA is recognized by TLR7 in pDC [100,101], a cell type in which HTLV-1 does not replicate, and this leads to a robust induction of IFN-I. In susceptible cells, reverse transcription in the presence of SAMHD1 generates viral DNA intermediates, such as the ssDNA90 [55], that were found to be sensed by IFI16 [102] and Ku70 [103]. ssDNA90 sensing by these DNA sensors then activates IFN-I production in a STING-dependent manner, which in turn dampens HTLV-1 productive infection [102,103]. As mentioned above, sensing of viral DNA intermediates might also lead to an abortive infection after the STING-dependent induction of apoptosis, as demonstrated in in vitro exposed monocytes [55].

Immature DC are permissive to HTLV-1 replication, although to a limited extend [6,7]. However, interestingly, no induction of IFN-I could be detected upon productive infection [8]. This suggests that in this cell type, viral sensing and / or antiviral signaling leading to IFN-I production is not efficient. How HTLV-1 might evade innate sensing in productively infected immature DC remains to be described. In contrast, autologous mature DC are resistant to infection [8], suggesting that maturation-induced restriction factors are still to be discovered. Because genes differentially expressed in immature *versus* mature DC are known [104], this offers a unique opportunity to screen for new HTLV-1 restriction factors in mature DC.

### 4.2. By Interrogating the Possible Relevance of Other Known Restriction Factors Active against HIV or Other Retroviruses

As reviewed in the previous sections, only a handful of HIV restriction factors have been tested against HTLV-1. In fact, many more HIV restriction factors are known, and the list has continued to expand in the last decade, as exemplified by the discovery of the anti-HIV properties of IFITM, MxB or SERINC3/5 [105,106,107,108,109], as well as those of other members of the TRIM family. A systematic analysis of their antiviral activity against HTLV-1 could therefore be of great interest, either to identify novel HTLV-1 restriction factors, should they prove active, or to help to pinpoint additional differences in HTLV-1 versus HIV cycles, should they prove inactive. Additional HIV restriction factors will probably be uncovered in the upcoming years, in particular by extending the search for new host species such as bats. In this regard, Morrison et al. recently identified that primate lentiviruses were restricted in *Pteropus alecto* cells [110]. This occurs through a new mechanism that remains to be fully described, and should also be tested against Deltaretroviruses such as HTLV-1.

Because HIV is the most studied of human retroviruses, much HTLV-1 research has been inspired by discoveries made in the HIV field. However, widening the scope to other retroviruses aside from HIV might also prove useful (for insights into new hints for anti-HTLV-1 factors, see Figure 3). Inputs from foamy viruses (FV) regarding their resistance to HIV resistance factors, such as A3G, SAMHD1 and Tetherin, have already been illustrated above. In addition, FV restriction factors have also been identified. For instance, the cellular Pirh2 protein (p53-induced RING-H2 protein) was shown to interact with the FV transactivator protein Tas, leading to Tas degradation and FV restriction [111]. Very recently, TRIM28, a TRIM family member already known to transcriptionally restrict retroviruses by direct binding to proviral sequences [112], was also described as a Tas-binding partner able to induce Tas proteasomal degradation [113]. Interestingly, Tax from HTLV-1 shares structural and functional properties with Tas [114]. The degradation of Tax by Pirh2 or TRIM28 could thus be an attractive hypothesis for novel mechanisms of HTLV-1 restriction.

Beyond retroviral restriction exerted by host-encoded proteins, restriction by endogenous retroviral element-encoded factors is also described. For instance, Refrex-1 is a soluble truncated Env protein from a feline endogenous retrovirus that was found to restrict infection by feline leukemia virus, a Gammaretrovirus. Interestingly, sequences from endogenous Deltaretroviruses have been detected in several mammals, including bats or dolphins [115,116,117]. Whether endogenous sequences of Deltaretroviruses are also present in primates, and whether they could control new rounds of exogenous Deltaretroviruses infection has not been investigated yet.

## 5. Conclusions

In conclusion, the identification of restriction factors active against HTLV-1 infection remains a challenging task. Interestingly, while HTLV-1 infection has been associated with several genetic polymorphisms in immune genes, these did not affect genes encoding restriction factors such as ISG [118], but rather genes encoding cytokines [119], suggesting that the pressure exerted by restriction might be relatively low, and targeted towards a reduction in cell proliferation rather than a reduction in cell infection. Of note, viral restriction by host factors is often balanced by viral factors that antagonize these restriction factors, or by the selection of escape variants, in which mutations allow resistance to restriction. This underlines the concept of the arms race between hosts and viruses [120], which translates into a constant balance between the selection of host strategies to restrict viral infection, and the selection of viruses that can overcome these restrictions. In stark contrast to HIV, HTLV-1 does not encode any viral protein whose primary function is to antagonize cell restriction factors, although this might also reflect our partial understanding of the functions of HTLV-1 regulatory and auxiliary proteins. In addition, while adaptive mutations in HIV and cellular genes encoding factors at the HIV / host interface have been observed, this is not the case in HTLV-1, whose genetic diversity is much lower than that of HIV [9,121]. This is first a consequence of the higher fidelity of HTLV-1 reverse transcriptase, whose mutation rate is four times lower than that of HIV, although similar to that of BLV [122]. Second, HTLV-1 also disseminates through clonal expansion of infected cells, which does not involve the viral reverse transcriptase (Figure 1).

In line with this last reflection, we should emphasize that HIV and HTLV-1 have evolved distinct modes of persistence in their host. While HIV persists through sustained viral production in vivo, HTLV-1 rather establishes a latent-like infection, with low (although continuous) levels of infectious spread in vivo that may be compensated by clonal expansion of infected cells. This could explain the distinct pressure exerted by restriction factors on HIV and HTLV-1. Interestingly, the clearest data on restriction factors active against HTLV-1 relate to factors targeting viral expression, such as CIITA and ZAP. In fact, these factors could be interpreted as beneficial for HTLV-1, as they enforce a low level of viral expression, which is most adapted to the mode of persistence of HTLV-1 [58]. This would also explain why no HTLV-1 escape mechanisms have evolved against these factors. Taken together, these observations underscore the limitation of relying on HIV as a reference to study HTLV-1.

Instead, searching for host factors that could inhibit HTLV-1 pro-proliferative action could represent an opportunity to identify HTLV-1-specific antiviral factors that could impede HTLV-1 clonal replication. Intriguingly, while HTLV-1 infects both CD4+ and CD8+ T cells and induces the clonal expansion of both cell types [35,123], the vast majority of ATL cases are CD4+ T cell leukemia, suggesting that some CD8+ T cell-specific factors could restrict the leukemogenic potential of HTLV-1 in this cell type. Accordingly, it has been shown that HTLV-1 induces excessive cell proliferation in CD4+ T cells, but not in CD8+ T cells, and that this is associated with the accumulation of cellular defects indicative of genetic instability in a CD4+-restricted manner [35]. Understanding how CD8+ T cells are refractory to HTLV-1 pro-proliferative action could be instrumental to identifying new sets of antiviral mechanisms adapted to viruses replicating via the clonal expansion of infected cells.

Primary CD4+ T cells isolated from infected individuals also show a heterogenous ability to undergo spontaneous proliferation ex vivo, which correlates with viral gene expression [124]. In addition to determinants linked to the integration site [125,126], identifying the host factors able to silence HTLV-1 expression will be of utmost interest to better understand whether host cells can escape HTLV-1-induced proliferation, at least in the early stages at which an “HTLV-1 memory” is not yet acquired, and thus design possible “block and lock” antiviral strategies [127]. Several host factors have been described as repressors of HTLV-1 transcription, such as Sp1 [128], FACT [129] or PRC1 [130], but their impact on HTLV-1-induced polyclonal proliferation remains to be investigated. Once more, investigating latency-inducing factors active against other retroviruses, such as the HUSH complex, which epigenetically silences the HIV genome (recently reviewed in [131]), could be a promising research direction.

In conclusion, while the classical concept of antiviral restriction against HTLV-1 might be rather inoperative in the chronic phase of infection, which shows low levels of infectious spread, the question of HTLV-1 restriction upon acute infection, which largely depends on infectious spread, remains open. Because DC are thought of as the first cells to become infected upon primo-infection, and because the infection of myeloid cells plays a crucial role in establishing HTLV-1 chronic infection [30,31,132], understanding the parameters that determine the efficiency of infection in DC specifically, and in myeloid cells more generally, is of utmost importance. In addition, efforts to pinpoint cell-type-, but also activation-state-specific restriction mechanisms, will definitely help to better characterize the replication cycle of HTLV-1 in myeloid cells, but also in T cells that are exposed to continuous, although mild, HTLV-1 infectious spread.

## Figures and Tables

**Figure 1 viruses-14-01611-f001:**
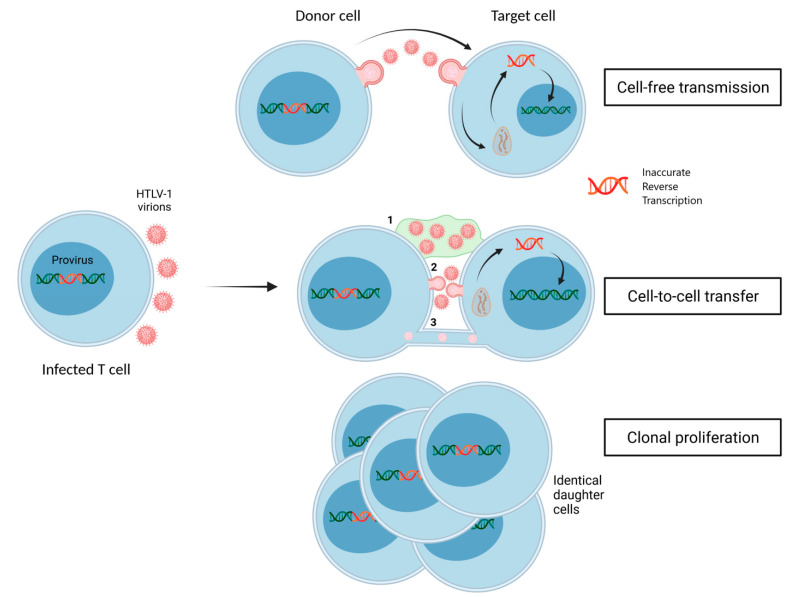
Modes of HTLV-1 dissemination. The donor cell is an infected CD4+ T cell containing a stably integrated provirus. In vitro cultures of infected cells produce low numbers of virions that are poorly infectious. Thus, cell-free transmission is not considered as an efficient mode of dissemination of HTLV-1, in contrast to other viruses such as HIV. Viral dissemination upon infectious spread instead occurs through cell-to-cell contacts using a biofilm (1), viral synapses (2) or protrusions and tunneling nanotubes (3). Following uncoating, the viral RNA is converted into a double-stranded viral DNA (represented in red) by the error-prone viral reverse transcriptase, thus introducing some genome variation. The viral DNA is then imported to the nucleus and integrated within the host genome. Since infected CD4+ T cells can undergo successive rounds of cell division, HTLV-1 is also disseminated through the clonal proliferation of infected cells, a process generating infected daughter cells sharing identical integration sites and identical viral sequences.

**Figure 2 viruses-14-01611-f002:**
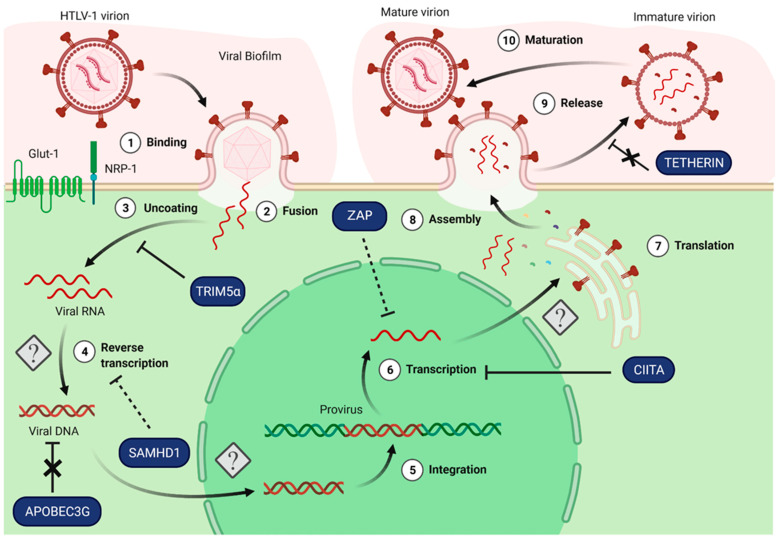
HTLV-1 viral replication cycle and restriction factors: knowledge and gaps. HTLV-1 viral replication cycle remains incompletely described, as indicated by the question marks. The model presented here relies on similarities with HIV viral replication cycle for steps remaining unresolved or not yet investigated. During early binding, HTLV-1 envelope proteins bind to NRP-1, Glut-1 and HSPG (not depicted) (1), which triggers conformational changes allowing fusion at the cell membrane (2). The two copies of the viral RNA genome are released in the cytoplasm (3) and converted into double-stranded DNA by the reverse transcriptase (4), although the exact sequence of events from uncoating to reverse transcription is not fully elucidated. After import of the viral dsDNA into the nucleus, a process that remains to be elucidated, viral DNA is integrated in the cellular genome by the viral integrase (5). During the late steps, the proviral genome is transcribed (6), viral mRNA are spliced and translated into viral proteins (7). These processes are not completely understood, but the HTLV-1 viral proteins Tax and Rex are known to control transcription and export of unspliced mRNA, respectively. Eventually, viral RNA and proteins assemble to form new viral particles (8) budding from the cell membrane and released as immature particles (9) that are later matured by viral protease cleavages (10). The antiviral activities of known restriction factors active against HIV (blue blocks) have been tested in the context of HTLV-1 infection. While some restriction factors such as TRIM5α and CIITA are able to restrict the HTLV-1 replication cycle to some extent, others are inefficient (APOBEC3G and Tetherin), or show a controversial activity on HTLV-1 (SAMHD1) or not fully described (ZAP) activity on HTLV-1.

**Figure 3 viruses-14-01611-f003:**
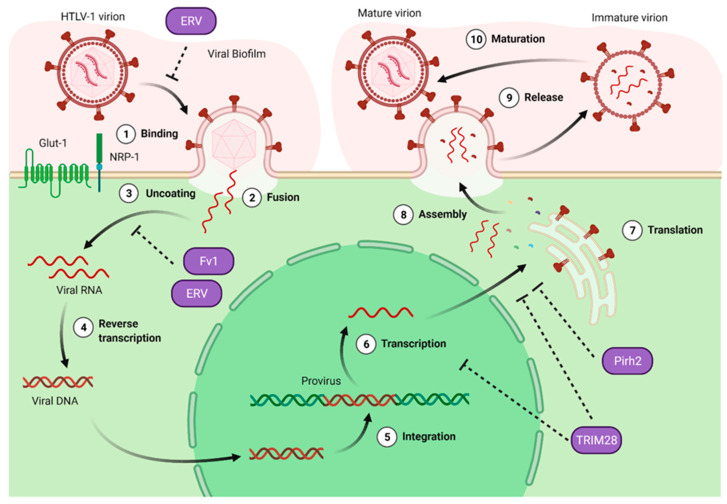
Hints for potential new factors able to restrict the HTLV-1 replication cycle. New HTLV-1 restriction factors can be searched by scrutinization of antiviral activities against viruses other than HIV. Among them, Pirh2 and TRIM28 target the Foamy virus transactivator Tas, leading to its degradation and restricting viral expression. Endogenous retroviral sequences (ERV) were found to restrict some exogenous retroviral infections in bats and feline animals, and they are thought to target the early steps of infection by inhibiting entry and uncoating. The uncoating step is also targeted by Fv1, a restriction factor identified in several retroviruses.

**Table 1 viruses-14-01611-t001:** Known restriction factors for HIV and their effects on HTLV-1.

Antiviral Factor	Mechanism of the Antiviral Effect on HTLV-1	References
TRIM family	TRIM5α may restrict HTLV-1 replication	[43,44]
	TRIM19/PML restricts HTLV-1 viral expression through Tax degradation	[45]
APOBEC3	No differences in A3G expression regarding the clinical status or the PVL	[46,47]
	Hypermutations of viral genomes indicating an evolutionary APOBEC3 editing in PBMC from asymptomatic carriers or ATL patients	[48,49,50]
	HTLV-1 is relatively resistant to A3G antiviral activity despite A3G being packaged into HTLV-1 virions	[48,51,52]
	A3A, A3B and A3H hapII are packaged into HTLV-1 virions and reduce viral replication in target cells	[53]
SAMHD1	No restriction by SAMHD1 in macrophages and dendritic cells	[54]
	Restriction of monocyte infection in vitro via SAMHD1-dependent abortive reverse transcription and induction of apoptosis	[55]
CIITA	CIITA restricts HTLV-1 viral expression through inhibition of Tax transactivating activity	[56,57]
ZAP	ZAP restricts HTLV-1 viral production, most probably by processing HTLV-1 transcripts	[58]
Tetherin	No restriction by Tetherin; in contrast, it may favor HTLV-1 spread	[59]

## Data Availability

Not applicable.

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
