# Peer review of "Is the HTLV-1 Retrovirus Targeted by Host Restriction Factors?"

_viruses, 2022, doi:10.3390/v14081611_

Round 1

Reviewer 1 Report

Carcone et al

This review by Carcone et al. provides a comprehensive overview on our sparse knowledge about restriction factors of HTLV-1. After a clear introduction into HTLV-1 replication and dissemination, the authors elaborate HTLV-1’s poor sensitivity to type-I interferon and summarize HTLV-1 studies that analysed the most prominent HIV restriction factors, including the TRIM family, APOBEC3G, SAMHD1, CIITA, ZAP, and Tetherin. At the end, the authors provide an outlook how to improve our knowledge on HTLV-1 restriction by getting better insights into HTLV-1 sensing and transposing the knowledge of restriction factors against other retroviruses than HIV. Overall, this manuscript is well-written and nicely illustrated. However, next to few minor comments, grammar and style should be revised as indicated below.

Minor comments:

-          Figure 1: The authors illustrate the routes of HTLV-1 cell-to-cell transmission. However, nanotubes should be renamed to „protrusive structures“ or „protrusions and tunneling nanotubes (TNTs)“ since „nanotubes“ may be misleading. Not only tunneling nanotubes, which are much thinner in size, but also protrusions (thicker in size) have been proposed to contribute to viral dissemination.

-          The authors use the term „viral cycle“ very frequently (e.g. line 115 and others), which is a rather uncommon term. I suggest to call it either „viral life cycle“ or viral replication cycle, depending on the context.

-          Line 265: It should be noted that Vpx is an HIV-2 encoded protein, which is not encoded by HIV-1.

Grammar/ style:

-          Please check grammar, e.g. line 46 „(…) the HTLV-1 genome is integrated (…)

-          „thus“ has to be followed by a comma (e.g. line 49 and many others)

-          Please check hyphenation, e.g. line 121 (re-verse), line 388 fac-tors, line 483 de-scribed, line 492 exert-ed, line 503 be-come

-          Line 131 „the“ is missing (to restrict the HTLV-1 viral …“; line 216: described steps oft he HTLV-1 viral …“

-          Table 1: typo „revesre“; Tetherin: „in contrast, it may favor“

-          Line 218: „might help to better understand“ instead of „might help better understand“

-          Line 279: „This thus“: please remove „thus“

-          Line 374: This sentence could be modified. Please include „in vitro“. „in the form of the viral biofilm“ could be replaced by „and concentrated in the viral biofilm“

-          Line 391: „might help to better understand“ instead of „might help better understand“

-          In line 409 the citation (ref. 46) is missing.

-          Line 428: „help to pinpoint“ instead of „help pinpoint“

-          Line 481: please check grammar of sentence, it is very difficult to understand.

-          Line 496: „HTLV-1’s mode“ instead of „HTLV-1 mode“

-          Line 508: „help to better characterize“ instead of „help better characterize“

Author Response

Thank you  for the positive comments and the deep reading of our manuscript. We have made  all changes requested and corrected accordingly the manuscript.

Please note that the manuscript has been updated to reply to reviewer#2 suggestions aiming at better comment the different selective pressures exherted by antiviral factors of the infected cells to prevent/control HIV or HTLV-1 infection. Accordingly we added the following paragraphs in the revised manuscript:

  • a discussion on HIV-1 and HTLV-1 different replication outcomes (i.e. release of new viral particle vs imortalization of infected cells respectively) in lines 88-93
  • a discussion on the mechanisms allowing HTLV-1 to induced persistent T cell clonal proliferation in line 196-222 and  references #34-42 that support the text
  • a discussion on new antiviral factors that would target HTLV-1-induced proliferation of infected cells in lines 533-556 and references #123-131 that support the text.

Reviewer 2 Report

In this review, the authors extensively discuss whether restriction factors in infected T cells discovered against HIV-1 restrict HTLV-1 infection, based on previous reports, as well as their own works. The article is appropriately organized by thoroughly reviewing former reports and provides a lot of useful information as a review of restriction factors for HIV-1. Also, the fact that the restriction factors of HIV-1 did not affect HTLV-1 as a whole provides an opportunity to reaffirm the differences in replication strategies between these two dissimilar viruses. If the purpose of this paper is to discuss the presence or absence of restriction factors in HTLV-1, however, I would like to suggest the following points to improve the importance of this review.

The conclusion that HIV-1 restriction factors do not have a similar effect on HTLV-1 is not totally unexpected. The authors mainly note the similarities in the replication cycle of HIV-1 and HTLV-1, thus hypothesizing that HIV-1 restriction factors may affect HTLV-1 replication efficiency. Indeed, the pathways from entry to release of both viruses can be outlined as shown in Figure 2. However, as the authors have pointed out several times in the text, HTLV-1 and HIV-1 have very different modes of infection and replication: HIV-1 has a strategy primarily on de novo infection by viral particles, while HTLV-1 replicates primarily by multiplication of infected cells. Although these viruses share CD4+ T cells as the cellular host, the molecular mechanisms of virus-host interaction after they invade CD4+ T cells can be quite different.

To argue about the presence or absence of host restriction factors specific for HTLV-1, they should rather focus on the differences between HIV-1 and HTLV-1.  For example, HIV-1 infection leads to massive particle release and infected T-cell death, whereas for HTLV-1, particle production is shut down at a certain level and infected cells are immortalized. Thus, to survive, highly efficient viral gene expression is essential for HIV-1, and the immortalization of infected cells is essential for HTLV-1. Although the detailed mechanism of immortalization of infected cells has not been elucidated, HTLV-1 may alter the cellular phenotypes by affecting the host epigenetic regulation and manipulating host gene expression patterns. Regulation of the AP-1 pathway by Tax and Hbz in the early stages of infection may also be involved in the regulation of infectious immunity around infected cells.

It is also an important point to discuss whether host T cells are under selection pressure to prepare strong restriction factors against HTLV-1. HIV-1 infection means the death of the host T cell since viral particle production is continued until the host cell death. On the other hand, HTLV-1 immortalizes the host T cell and allows it to survive for decades. Accordingly, it can be expected that the selection pressure of host cell restriction factors has been very strong for HIV-1, but not for HTLV-1. Indeed, the authors mention that there are several HIV-1 restriction factors in host T cells, and HIV-1 accessory proteins also function to antagonize these host factors, whereas no clear host restriction factors have been found for HTLV-1. Such weak host cell restriction against HTLV-1 may explain why HTLV-1 has survived for tens of thousands of years without being eliminated from the human life cycle.

The theme of this review is one of the critical aspects of host-pathogen interactions. To increase the importance of this review, I would like to request the authors to expand the discussion on the molecular mechanism of the HTLV-1-specific mode of replication and the possible host restriction factors, which may limit HTLV-1-induced host cell immortalization, survival, and proliferation.  Expanding the scope of the search for restriction factors against HTLV-1 will lead to a deeper understanding of its replication cycle, which has not been fully understood. On top of that, comparison with HIV-1 will also improve our knowledge of the host-virus co-evolution mediated by host restriction factors and viral factors.

Author Response

In this review, the authors extensively discuss whether restriction factors in infected T cells discovered against HIV-1 restrict HTLV-1 infection, based on previous reports, as well as their own works. The article is appropriately organized by thoroughly reviewing former reports and provides a lot of useful information as a review of restriction factors for HIV-1. Also, the fact that the restriction factors of HIV-1 did not affect HTLV-1 as a whole provides an opportunity to reaffirm the differences in replication strategies between these two dissimilar viruses. If the purpose of this paper is to discuss the presence or absence of restriction factors in HTLV-1, however, I would like to suggest the following points to improve the importance of this review.

The conclusion that HIV-1 restriction factors do not have a similar effect on HTLV-1 is not totally unexpected. The authors mainly note the similarities in the replication cycle of HIV-1 and HTLV-1, thus hypothesizing that HIV-1 restriction factors may affect HTLV-1 replication efficiency. Indeed, the pathways from entry to release of both viruses can be outlined as shown in Figure 2. However, as the authors have pointed out several times in the text, HTLV-1 and HIV-1 have very different modes of infection and replication: HIV-1 has a strategy primarily on de novo infection by viral particles, while HTLV-1 replicates primarily by multiplication of infected cells. Although these viruses share CD4+ T cells as the cellular host, the molecular mechanisms of virus-host interaction after they invade CD4+ T cells can be quite different.

Reply: We thank the reviewer for their comments. We acknowledge that the discussion on the different outcomes of HIV-1 and HTLV-1 infection was not enough detailed. We have now included a discussion on the different replication strategies deployed by HIV-1 and HTLV-1 that might reflect their different susceptibility to selection pressure exerted by the host. This is found in lines 88-93 of the revised manuscript.

To argue about the presence or absence of host restriction factors specific for HTLV-1, they should rather focus on the differences between HIV-1 and HTLV-1.  For example, HIV-1 infection leads to massive particle release and infected T-cell death, whereas for HTLV-1, particle production is shut down at a certain level and infected cells are immortalized. Thus, to survive, highly efficient viral gene expression is essential for HIV-1, and the immortalization of infected cells is essential for HTLV-1. Although the detailed mechanism of immortalization of infected cells has not been elucidated, HTLV-1 may alter the cellular phenotypes by affecting the host epigenetic regulation and manipulating host gene expression patterns. Regulation of the AP-1 pathway by Tax and Hbz in the early stages of infection may also be involved in the regulation of infectious immunity around infected cells.

Reply: We have now included a discussion (lines 196-222 of the revised manuscript) on how HTLV-1 induces persistent T cell clonal expansion, through the action of the viral proteins Tax and HBZ on the cell cycle, that favors subsequent genetic and epigenetic modifications of the host genome and lead to immortalization and transformation. Accordingly, the references #34-42 of the revised manuscript have been included to support the argumentation.

It is also an important point to discuss whether host T cells are under selection pressure to prepare strong restriction factors against HTLV-1. HIV-1 infection means the death of the host T cell since viral particle production is continued until the host cell death. On the other hand, HTLV-1 immortalizes the host T cell and allows it to survive for decades. Accordingly, it can be expected that the selection pressure of host cell restriction factors has been very strong for HIV-1, but not for HTLV-1. Indeed, the authors mention that there are several HIV-1 restriction factors in host T cells, and HIV-1 accessory proteins also function to antagonize these host factors, whereas no clear host restriction factors have been found for HTLV-1. Such weak host cell restriction against HTLV-1 may explain why HTLV-1 has survived for tens of thousands of years without being eliminated from the human life cycle.

The theme of this review is one of the critical aspects of host-pathogen interactions. To increase the importance of this review, I would like to request the authors to expand the discussion on the molecular mechanism of the HTLV-1-specific mode of replication and the possible host restriction factors, which may limit HTLV-1-induced host cell immortalization, survival, and proliferation.  Expanding the scope of the search for restriction factors against HTLV-1 will lead to a deeper understanding of its replication cycle, which has not been fully understood. On top of that, comparison with HIV-1 will also improve our knowledge of the host-virus co-evolution mediated by host restriction factors and viral factors.

Reply: We thank the reviewer for their comments. To address this specific comment that we indeed believe is very important, we included a new paragraph in the conclusion (lines 533-556 of the revised manuscript) to propose the pro-proliferative action of HTLV-1-infected cells as a new target for the identification of HTLV-1-specific antiviral factors. We also discussed the role of some repressors of HTLV-1 transcription as putative “block and lock” antiviral strategies. To support this discussion we added the references #123-131 in the revised manuscript.

Round 2

Reviewer 2 Report

Thank you for the comprehensive revisions in response to all of my comments.

I would like to request a minor change in L.199-202 and L.553 regarding “clonal expansion” of HTLV-1 infected T cells in asymptomatic carriers, HAM/TSP, and ATL patients. As you mentioned in L. 545, HTLV-1 infected cells from infected individuals are quite heterologous in terms of clonality. During latency, each clone of HTLV-1 infected cells proliferates time-to-time (still “probably”, not confirmed clinically, yet) to maintain a certain number of infected cells in asymptomatic carriers. Even in HAM/TSP patients and indolent ATL patients, the number of infected cells and malignant cells increases in a polyclonal or oligoclonal manner. Mono-clonal expansion only occurs in ATL-transformed cells in acute-type ATL patients.  Thus, “clonal-expansion” in these sentences should be changed to “(polyclonal) infected cell proliferation”. 

Author Response

Thank you again your deep editing and precise wording of our manuscript. We made the 2 suggested changes in line 199 and line 553.

This manuscript is a resubmission of an earlier submission. The following is a list of the peer review reports and author responses from that submission.

Round 1

Reviewer 1 Report

This review by Carcone et al. provides a comprehensive and state-of-the art overview on the role of known restriction factors of HIV during replication of the related retrovirus HTLV-1. After a clear introduction into HTLV-1 replication and pathogenesis, the authors elaborate HTLV-1’s poor sensitivity to type-I interferon and summarize HTLV-1 studies that analysed the most prominent HIV restriction factors, including the TRIM family, APOBEC3G, SAMHD1, CIITA, and Tetherin. At the end, the authors provide an outlook how to improve our knowledge on HTLV-1 restriction by getting better insights into HTLV-1 sensing and transposing the knowledge of restriction factors against other retroviruses than HIV. Overall, this manuscript is very well-written and nicely illustrated.

Minor comments:

  • Line 35: „is maintained“
  • Line 91 „envelope“ instead of „envelop“
  • Figure 2/ line 106: Authors describe that viral particle produced can be released in the extracellular medium. This is true in cell culture, however, there is no evidence for cell-free particles in vivo. Therefore, the authors should specify this in more detail in the Figure and the corresponding text.
  • The authors list several restriction factors including APOBEC3G. Maybe the authors could expand their list and include another study that compared A3 proteins in HTLV-1 reporter assays and found that A3A, A3B, and A3H haplotype 2 act as inhibitors of HTLV-1 (Ooms et al., J Virol 2012).

Reviewer 2 Report

This is a review intended to highlight and contrast current knowledge regarding the HIV “restriction factors” and other antiviral host cell mechanisms/factors reported in the literature on innate immune responses or restrictions against HTLV-1 replication. While the topic is of interest to readers, The manuscript appears to be prepared by a person who is new to the field. First and foremost, viral restriction factors and general host innate immune responses (such as type 1 interferons and cytosolic nucleic acid sensors) are not clearly delineated in the review.  The various HIV restriction factors that antagonize specific steps in HIV replication were countered/ inhibited by virally encoded proteins such as Vif, Vpr, Vpu, and Vpx. As summarized in the manuscript, most if not all HIV restriction factors appear to have little to no effect on blocking HTLV replication except perhaps APOBEC3G, which is inhibited by HTLV-1 NC. While some of the cited anti-HTLV-1 factors such as CIITA and TRIM19 may dampen Tax activities or destabilize Tax, they are really not “restriction” factors per se as no HTLV-1-encoded factors directly inactivate them. Furthermore,  the authors appear to be unaccustomed to communicating in English. As such, the manuscript in its present form is not ready for prime time and needs substantial improvement in both the scientific contents, especially the accuracy of the information and the scientific nomenclature used (see specific comments below) and the English writing (some of them indicated by highlights and strikethroughs, too numerous to cite individually).

Comments:

In Fig. 2, the authors appear to be oblivious of the published literature on the “virological synapse” as the major conduit for HTLV-1 transmission, and mentioned only the viral biofilm as the means by which HTLV-1 infection occurs.

HTLV-1 viral reverse transcriptase is only 5x less error-prone than other retroviral RT, and should not be characterized as a highly accurate RT. The relative lack of viral genetic heterogeneity is largely a result of its replicative strategy, which entails establishment of viral latency and sporadic and limited reactivation for de novo infection. 

Rex should not be described as a protein that affect viral mRNA splicing. Its primary function like that of the HIV Rev is to facilitate transport of unspliced and singly splices viral mRNAs to the cytosol.

In line 195, APOBEC3G was reported to be inhibited by HTLV-1 nucleocapsid protein and should be described as such.

It is not clear what the authors meant by “intrinsic” antiviral response or immunity? If innate immunity is being referred to, then state so correctly. Otherwise, define “intrinsic immunity” vis-à-vis “innate immunity” clearly.

Are the authors referring to viral uncoating or viral disassembly by the term “decapsidation”?
